# Marine reservoir ages for coastal West Africa

Guillaume Soulet[1], Philippe Maestrati[2], Serge Gofas[2,3], Germain Bayon[1], Fabien Dewilde[1], Maylis Labonne[4], Bernard Dennielou[1], Franck Ferraton[4], Giuseppe Siani[5]

1 Ifremer, Univ Brest, CNRS, Geo-Ocean UMR6538, F-29280, Plouzané, France
2 Muséum National d'Histoire Naturelle, Paris, DGD-Collections, France
3 Departamento de Biología Animal, Facultad de Ciencias, Universidad de Málaga, Málaga, Spain
4 MARBEC, Univ Montpellier, CNRS, Ifremer, IRD, Montpellier, France
5 GEOPS, UMR 8148 Université Paris-Saclay, Orsay, France

*Correspondence to*: Guillaume Soulet (gsoulet@ifremer.fr)

**Abstract.** We measured the $^{14}$C age of pre-bomb suspension-feeding bivalves of known-age from coastal West Africa (n=30) across a latitudinal transect extending from 33°N to 15°S. The specimens are from the collections of the Muséum National d'Histoire Naturelle (Paris, France). They were carefully chosen to ensure that the specimens were alive when collected or died not long before collection. From the $^{14}$C-dating of these know-age bivalves, we calculated the marine reservoir age (as ΔR and R values) for each specimen. ΔR values were calculated relative to the Marine20 calibration curve and the R values relative to Intcal20 or SHcal20 calibration curves. Except five outliers, the ΔR and R values were quite homogenous to a mean value of -77 ± 47 $^{14}$C yrs (1sd, n = 25), and of 400 ± 59 $^{14}$C yrs (1sd, n = 25), respectively. These values are typical of low latitude marine reservoir age values. Five suspension-feeding species living in five different ecological habitats were studied. For localities were different species were available, the results yielded similar results whatever the specie considered suggesting that the habitat has only a limited impact on the marine reservoir age reconstruction. We show that our measured marine reservoir ages follow the declining trend of the global marine reservoir age starting ca. 1900 AD, suggesting that marine reservoir age of coastal West Africa is driven, at least at first order, by the global carbon cycle and climate rather than by local effects. Each outlier was discussed. Sub-fossil specimens likely explain the older $^{14}$C age and thus larger marine reservoir age measured for these samples. *Bucardium ringens* might not a best choice for marine reservoir age reconstructions.

## 1 Introduction

The marine reservoir age (R) at a given calendar time/year (t) is the difference between the radiocarbon age ($^{14}$C) of the dissolved inorganic carbon (DIC) of the ocean ($^{14}C_m$) and that of atmospheric $CO_2$ ($^{14}C_{atm}$) (Stuiver and Polach, 1977; Ascough et al., 2005; Soulet et al., 2016; Skinner and Bard, 2022; Soulet, 2015):

$$R(t) = {}^{14}C_m(t) - {}^{14}C_{atm}(t) \quad (1)$$

At global scale, the marine reservoir age of the surface mixed layer of the ocean is set by the exchange of "young" $CO_2$ at the atmosphere-ocean interface, plus the exchange of DIC between oceanic surface waters and deep waters that contain large

amounts of "old" DIC (Bard, 1988; Skinner and Bard, 2022). The $^{14}$C age of the global ocean over time, i.e., the Marine20 calibration curve (Heaton et al., 2020), has been modelled using the global carbon cycle box model BICYCLE (Köhler et al., 2006; Köhler and Fischer, 2006, 2004; Köhler et al., 2005) and the Northern Hemisphere atmospheric $^{14}$C calibration curve (IntCal20; Reimer et al., 2020). While the global marine calibration curve (Marine20) is widely used to derive calibrated ages from $^{14}$C dating of marine samples, it does not account for local marine $^{14}$C offsets due to, for instance, continental carbon inputs to the coastal ocean, regional winds, and changes in the oceanic circulation and climate (Bard, 1988; Alves et al., 2018; Skinner and Bard, 2022; Heaton et al., 2023). Hence, the usefulness of the ΔR metric (Stuiver et al., 1986; Stuiver and Braziunas, 1993; Reimer and Reimer, 2017) that is the difference between the $^{14}$C age of any marine sample ($^{14}C_m$) and that of the marine calibration curve ($^{14}C_{Marine20}$) at the same time (t):

$$\Delta R(t) = {}^{14}C_m(t) - {}^{14}C_{Marine20}(t) \quad (2)$$

The local marine reservoir age offset (ΔR) is known to vary largely as demonstrated by pre-bomb values ranging between – 500 to + 2000 $^{14}$C years (Reimer and Reimer, 2001) depending on the location. Larger ΔR values are located at high-latitudes while values close to ΔR = 0 $^{14}$C years are located at low latitudes (Bard, 1988; Bard et al., 1994).

From a geochronological perspective (i.e., calibration of marine $^{14}$C dates and building age-depth model from marine $^{14}$C dates), knowing ΔR(t) is of crucial interest to correct marine $^{14}$C dates for a local $^{14}$C offset compared to the global marine calibration curve and hence a pre-requirement to derive accurate calendar ages. Reconstructing ΔR(t) values from unstudied areas is also valuable as it could contribute to deriving regional/local marine calibration curves from the global one using 3-D large-scale ocean circulation model (Butzin et al., 2017; Alves et al., 2019).

From a carbon cycle perspective, the R(t) and ΔR(t) are also very important as they reflect $^{14}$C disequilibria between the ocean and the atmosphere and hence they are key proxies to understand local variations of the global carbon cycle, and its evolution over time with changing climate and environment (Skinner et al., 2015, 2010; Lindsay et al., 2016; Soulet et al., 2011; Siani et al., 2001; Schefuß et al., 2016; Heaton et al., 2021).

On a whole, efforts to estimate R(t) and ΔR(t) values wherever possible are valuable to the understanding of both modern and past carbon cycle, and the reconstruction of climate and environmental changes based on sedimentary archives.

Pre-bomb R(t) and ΔR(t) values for coastal West Africa are very sparse. According to the Marine Reservoir Correction Database (Reimer and Reimer, 2001; http://calib.org/marine/; last seen 15/11/2022), from Oran on the Mediterranean coast of Algeria (Siani et al., 2000) to Hondelkip Bay on the Atlantic coast of South Africa (Dewar et al., 2012), only few marine reservoir ages from Mauritania and Senegal were reported (Ndeye, 2008) (Fig. 1). For Mauritania, collection sites were Nouadhibou (formerly Port-Étienne, 2 samples), the area of the Cape Timiris – El Mamghar (3 samples). Two samples were collected from an unknown location from coastal Mauritania. For Senegal, collection sites were restricted to the Dakar area (Almadies, Dakar harbour, Gorée Island and Rufisque; 5 samples). Thirteen additional samples were from unknown locations from coastal Senegal.

In this study we report new marine reservoir age values (n=30) based on the $^{14}$C dating of bivalves with a known pre-bomb collection date and collected across a latitudinal transect extending from Mohammedia (Morocco, 33°N) to Moçâmedes

(Angola, 15°S). Our suite of sample includes specimens from Mauritania, Senegal, Republic of Guinea, Sierra Leone, Ivory Coast, Benin, Gabon and Republic of Congo (Fig. 1, Table S1 in the Supplement). We used specimens of five different species: *Senilia senilis*, *Bucardium ringens*, *Donax rugosus*, *Ostrea stentina* and *Pseudochama gryphina*. We briefly discuss our results in the context of local environmental setting of the studied bivalves and regional oceanography of the Eastern Atlantic Ocean.

## 2 Material and methods

### 2.1 Material

Bivalve shells were selected from the collections of the Muséum National d'Histoire Naturelle (MNHN) (Paris, France). We carefully chose pre-bomb specimens of known-age and ensured that they were collected alive or very soon after death. For example, specimens with articulated valves and exhibiting flesh remains inside the shell were clearly collected alive. For *Senilia senilis*, the presence of the fragile periostracum provides evidence that the specimen was collected fresh. For *Bucardium ringens*, remains of the hinge ligament indicate that the bivalve death occurred not long before collection. The collection date was also carefully checked. Below, we provide background information for the five different bivalve species investigated in this study. Additional information for each sample is given in section 3.1.

*Senilia senilis* (Linnaeus, 1758) can be found from Mauritania to northern Angola. It lives in fine sand in estuaries, creeks or lagoon with regular tidal influence from the lower intertidal zone to about 2 meters water depth. The specie is tolerant to seasonal salinity changes (von Cosel and Gofas, 2019). *S. senilis* is a suspension feeder that lives in the top 5-10-cm layer of the sediment (Okera, 1976; Catry et al., 2017).

*Bucardium ringens* (Bruguière, 1789) occurs from Mauritania to southern Angola. It lives in clean fine sand and mixed sand on open coast from shallow (5-10 meters depth) to about 50 meters depth. Shells and valves are commonly cast ashore on beaches but live-taken specimens are rare (von Cosel and Gofas, 2019). *B. ringens* is likely a suspension feeder as typically are cardiids (Herrera et al., 2015).

*Donax rugosus* (Linnaeus, 1758) occurs from Mauritania to Ghana and from northern Angola to southern Angola. It lives in mixed and coarse sand in the surf zone of open beaches (von Cosel and Gofas, 2019). *D. rugosus* is a suspension feeder (Smith, 1971).

*Ostrea stentina* (Payraudeau, 1826) can be found southern Portugal to Ghana, then from Gabon to northern Angola. It is common and occurs on various types of hard substrate such as rocks, stones, pebbles and other oysters from 1 to 30 meters depths. It can also be found in lagoons, inlets and creeks under marine condition (von Cosel and Gofas, 2019). *O. stentina* is a suspension feeder (Türkmen et al., 2005).

*Pseudochama gryphina* (Lamarck, 1819) occurs from Southern Portugal to Mauritania and from Gabon to southern Angola (von Cosel and Gofas, 2019) and lives on hard substrate such as rocks and stones in clear water offshore about 10 to 60 meters water depth. *P. gryphina* is a suspension feeder (Sessa et al., 2013).

A small piece (30-100 mg) of the outermost layers of each shell was cut using a Dremel™ rotary tool fitted with a cut-off wheel. We focused on the external part of the shell to ensure that we sampled and dated the most recent part (likely the last few months) of the specimen. The shell carbonate samples were then sonicated and rinsed in deionized water at least 5 times. Samples were coarsely crushed and split into a subsample for stable isotopic analysis and a subsample for $^{14}C$ analysis.

### 2.2 Radiocarbon measurements

Samples were washed with dilute $HNO_3$ (0.01M) for 15 min then rinsed to neutral pH. Then the shell carbonate was converted into $CO_2$ following LMC14 laboratory (Laboratoire de Mesure du Radiocarbone, Saclay, France) standard phosphoric acid hydrolysis procedure (Tisnérat-Laborde et al., 2001; Dumoulin et al., 2017). The $CO_2$ was then converted to graphite (Cottereau et al., 2007; Dumoulin et al., 2017) and analyzed for its $^{14}C$ composition by Accelerator Mass Spectrometry (AMS) using the Artémis $^{14}C$ AMS facility (Moreau et al., 2013). Results are corrected for the $^{13}C/^{12}C$ ratio as measured on the AMS (Santos et al., 2007) and are reported in the $F^{14}C$ notation (Reimer et al., 2004). $F^{14}C$ is identical to the $A_{SN}/A_{ON}$ metric (Stuiver and Polach, 1977), and the $^{14}a_N$ notation (Mook and van der Plicht, 1999). Corresponding conventional $^{14}C$ ages reported in $^{14}C$ years Before Present (AD 1950) were calculated according to:

$$^{14}C = -8033\ln(F^{14}C) \quad (3)$$

### 2.3 Stable carbon isotopes

Stable carbon and oxygen isotopic analyses of the dated samples were performed at the Pôle Spectrométrie Océan (PSO, Plouzané, France) using a MAT-253 (Thermo Scientific) stable isotope ratio mass spectrometer (IRMS) coupled with a Kiel IV Carbonate Device (Thermo Scientific). The measurements are reported versus Vienna Pee Dee Belemnite standard (VPDB) defined with respect to two international carbonate standards: NBS-19 ($\delta^{18}O$ = -2.20 ‰ and $\delta^{13}C$ = +1.95 ‰) and NBS-18 ($\delta^{18}O$ = -23.20 ‰ and $\delta^{13}C$ = -5.01 ‰). The mean external reproducibilities (1σ), based on repeated measurements of an in-house standard, was ±0.04‰ and ±0.02‰ for $\delta^{18}O$ and $\delta^{13}C$ values, respectively.

### 2.4 Marine Reservoir Age calculation

The marine reservoir age R of the selected shells is calculated according to equation (1) where t is the collection year as known from the museum records (Table S1 in the Supplement and section results), $^{14}C_m$ is the measured shell $^{14}C$ age, and $^{14}C_{atm}$ is the $^{14}C$ age of the atmosphere. For shells picked from the northern hemisphere, $^{14}C_{atm}$ is obtained from the IntCal20 calibration curve (Reimer et al., 2020). For shells from the southern hemisphere, we used instead the southern hemisphere calibration curve SHCal20 (Hogg et al., 2020). The uncertainty is calculated (Soulet, 2015) according to:

$$\sigma_{R(t)} = \sqrt{\sigma^2_{^{14}C_m(t)} + \sigma^2_{^{14}C_{atm}(t)}} \quad (4)$$

Note that mean SHCal20 offset compared to IntCal20 is estimated to be 36 ± 27 $^{14}C$ yrs. Thus, the R values calculated with IntCal20 or SHcal20 are essentially the same if one take uncertainties into account.

The local marine reservoir offset ΔR of the selected shells is calculated according to equation (2) where t is the collection year as known from the museum records (Table S1 in the Supplement and section results), $^{14}C_m$ is the measured shell $^{14}C$ age, and $^{14}C_{Marine20}$ is the $^{14}C$ age of the global marine calibration curve. The uncertainty is calculated as follows:

$$\sigma_{\Delta R(t)} = \sqrt{\sigma^2_{^{14}C_m(t)} + \sigma^2_{^{14}C_{Marine20}(t)}} \qquad (5)$$

Note that Reimer and Reimer (2017) do not propagate the uncertainty of Marine20 calibration curve.

## 3 Results and Discussion

### 3.1 Radiocarbon measurements results

The results are shown in Table S1 in the Supplement. Below we provide the detailed list of the samples used in this study. We classified the samples by location with corresponding geographic coordinates, then by species. Code numbers “MNHN-IM-2022-xxxx” allows one to find the samples in the collections of the MNHN of Paris (France). Code numbers “SacA-xxxxx” are the radiocarbon laboratory number for the sample.

#### 3.1.1 Samples from Morocco

MNHN-IM-2022-4615
*Ostrea stentina*
Mohammedia (33.71°N, 7.37°W)
Articulated specimen with remains of flesh.
SacA-68834: 482 ± 18 BP
$F^{14}C = 0.9418 \pm 0.0021$
$\delta^{13}C = 1.19$ ‰ VPDB
Collection date: 1921
Collector: Jacques de Lépiney
Museum label: Ostrea stentina Payr, Fedhala 1921, donat J de Lépiney 1939

MNHN-IM-2022-4609
*Ostrea stentina*
El Jadida, beach (33.25°N, 8.49°W)
An isolated valve looking quite fresh.
SacA-68828: 817 ± 18 BP
$F^{14}C = 0.9033 \pm 0.0020$
$\delta^{13}C = 0.71$ ‰ VPDB

Collection date: October 26th 1909

Collector: Louis Gentil

Museum label: Plage de Mazagan 26 octobre 1909, Maroc, Louis Gentil

MNHN-IM-2022-4608

*Ostrea stentina*

Lagoon of Sidi Moussa (32.98°N, 8.75°W)

Articulated specimen with remains of flesh.

SacA-68827: 1058 ± 19 BP

$F^{14}C = 0.8766 \pm 0.0021$

$\delta^{13}C = -0.48$ ‰ VPDB

Collection date: 1924

Collector: Jacques de Lépiney

Museum label: Ostrea stentina Payr., lagune de Sidi Moussa (région de Mazagan), 1924, donat. J de Lépiney 1939

**3.1.2 Samples from Mauritania**

MNHN-IM-2022-4612

*Ostrea stentina*

Nouadhibou, Pointe Chacal (20.91°N, 17.04°W)

Articulated specimen.

SacA-68831: 514 ± 17 BP

$F^{14}C = 0.9381 \pm 0.0020$

$\delta^{13}C = 2.28$ ‰ VPDB

Collection date: 1948

Collector: Roger Sourie

Museum label: Port Etienne (Pointe des Chacals) M. Sourie, 1948

MNHN-IM-2022-4610

*Ostrea stentina*

Cansado Bay (20.88°N, 17.04°W)

Articulated specimen with remains of flesh.

SacA-68829: 516 ± 17 BP

$F^{14}C = 0.9378 \pm 0.0020$

$\delta^{13}C$ = 2.09 ‰ VPDB

Collection date: 1911-1912

Collector: Mission Gruvel

Museum label: Ostrea stentina Payr. = lacerans Hanl., Baie de Cansado, Mission Gruvel, 1911-1912

MNHN-IM-2022-4599

*Bucardium ringens*

Nouadhibou (20.88°N, 17.04°W)

An isolated valve of a juvenile with remains of the hinge ligament.

SacA-68811: 863 ± 18 BP

$F^{14}C$ = 0.8981 ± 0.0020

$\delta^{13}C$ = 0.38 ‰ VPDB

Collection date: 1908

Collector: Mission Gruvel

Museum label: Cardium ringens Gmelin; Port Etienne; 1908; Mission Gruvel

MNHN-IM-2022-4603

*Donax rugosus*

Ndiago, beach (16.17°N, 16.51°W)

Articulated specimen with remains of flesh and hinge ligament.

SacA-68815: 518 ± 19 BP

$F^{14}C$ = 0.9376 ± 0.0022

$\delta^{13}C$ = 0.68 ‰ VPDB

Collection date: January 21st 1908

Collector: Mission Gruvel

Museum label: Donax rugosus Linné, N'Diago plage, 21.I.08, Mission Gruvel

### 3.1.3 Samples from Senegal

MNHN-IM-2022-4607

*Donax rugosus*

Saint Louis (16.02°N, 16.51°W)

Articulated specimen with remains of flesh and hinge ligament.

SacA-68819: 574 ± 18 BP

$F^{14}C$ = 0.9310 ± 0.0021

$\delta^{13}C$ = 1.29 ‰ VPDB

Collection date: December 1901

Collector: Buchet's mission

Museum label: Sénégal, Saint Louis, Coquilles; Donax, M$^{on}$ Buchet, X$^{bre}$ 1901

MNHN-IM-2022-4592

*Senilia senilis*

Dakar, backwaters of the "Marigot de Hann" (14.74°N, 17.39°W)

Articulated specimen with remains of flesh and well-preserved periostracum.

SacA-68824: 560 ± 17 BP

$F^{14}C$ = 0.9327 ± 0.0020

$\delta^{13}C$ = -0.29 ‰ VPDB

Collection date: May 1908

Collector: Mission Gruvel

Museum label: Arca (Senilia) senilis Linné, Marigot de Hann V.1908, se vend sur le marché de Dakar env. 2 sous la douzaine, Mission Gruvel

Note: The Marigot of Hann seems to have been a creek more or less connected to the ocean. It was drawn on a map of Dakar in 1905 but does not exist any longer. The map can be accessed from the Gallica website managed by the Bibliothèque Nationale de France: https://gallica.bnf.fr/ark:/12148/btv1b53197802m

MNHN-IM-2022-4593

*Senilia senilis*

Dakar, Bay of Hann, Pointe Bel Air, beach at low tide (14.71°N, 17.42°W)

Articulated specimen with remains of flesh and well-preserved periostracum.

SacA-68825: 544 ± 18 BP

$F^{14}C$ = 0.9346 ± 0.0020

$\delta^{13}C$ = 0.07 ‰ VPDB

Collection date: December 1$^{st}$ 1909

Collector: Mission Gruvel

Museum label: Arca (Senilia) senilis Linné, Baie de Hann, Pointe de Bel Air, plage à basse mer, M Gruvel, 1.XII.1909

MNHN-IM-2022-4606

*Donax rugosus*

Dakar, Bay of Hann (14.71°N, 17.42°W)

Articulated specimen with remains of flesh and hinge ligament.
SacA-68818: 526 ± 18 BP
$F^{14}C = 0.9366 \pm 0.0022$
$\delta^{13}C = 0.33$ ‰ VPDB
Collection date: April 1908
Collector: Mission Gruvel
Museum label: Donax rugosus Linné, Baie de Hann à basse mer, IV 08, Mission Gruvel

MNHN-IM-2022-4598
*Bucardium ringens*
Dakar, Bay of Hann at low tide (14.71°N, 17.42°W)
An isolated valve with remains of the hinge ligament.
SacA-68810: 628 ± 17 BP
$F^{14}C = 0.9247 \pm 0.0019$
$\delta^{13}C = 0.68$ ‰ VPDB
Collection date: April 1908
Collector: Mission Gruvel
Museum label: Cardium ringens Gmelin, Baie de Hann à basse mer, IV.08, Mission Gruvel

MNHN-IM-2022-4616
*Ostrea stentina*
Dakar, beach of Hann, posts of the pontoon (14.71°N, 17.42°W)
An isolated valve with remains of flesh
SacA-68835: 539 ± 19 BP
$F^{14}C = 0.9351 \pm 0.0022$
$\delta^{13}C = 1.56$ ‰ VPDB
Collection date: 1947
Collector: Roger Sourie
Museum label: Ostrea stentina Payr Dakar (plage de Hann, piles du ponton) M Sourie 1947

#### 3.1.4 Samples from Republic of Guinea

MNHN-IM-2022-4601
*Bucardium ringens*
Los Islands, Roume Island at low tide (9.46°N, 13.79°W)

A fresh-looking isolated valve.

SacA-68813: 857 ± 39 BP

$F^{14}C = 0.8988 \pm 0.0043$

$\delta^{13}C = -0.40$ ‰ VPDB

Collection date: December 20th 1909

Collector: Mission Gruvel

Museum label: Cardium ringens Gmelin; Ile Roumé, archipel de Los, à basse mer, 20.XII.09, Mission Gruvel

MNHN-IM-2022-4618

*Pseudochama gryphina*

Los Islands, Tamara Island (9.46°N, 13.83°W)

An articulated specimen.

SacA-68820: 502 ± 19 BP

$F^{14}C = 0.9395 \pm 0.0022$

$\delta^{13}C = 1.52$ ‰ VPDB

Collection date: 1909-1910

Collector: Mission Gruvel

Museum label: Chama gryphina Lm, Tamara, Guinée, mission Gruvel, 1909-1910

Note: It is possible that this sample was also collected in December 1909 as sample MNHN-IM-2022-4601

**3.1.5 Sample from Sierra Leone**

MNHN-IM-2022-4611

*Ostrea stentina*

Near Cape Saint Ann (7.56°N, 12.94°W)

Articulated specimen with remains of flesh.

SacA-68830: 464 ± 18 BP

$F^{14}C = 0.9439 \pm 0.0021$

$\delta^{13}C = 1.04$ ‰ VPDB

Collection date: 1912

Collector: Mission Gruvel

Museum label: Sierra Léone près Cap Ste Anne, m. Gruvel, 1912

**3.1.6 Samples from Benin**

MNHN-IM-2022-4591

*Senilia senilis*

Ahémé Lake (6.42°N, 1.96°E)

Articulated specimen with well-preserved periostracum

SacA-68823: 414 ± 18 BP

$F^{14}C = 0.9498 \pm 0.0021$

$\delta^{13}C = -4.76$ ‰ VPDB

Collection date: February 1910

Collector: Mission Gruvel

Museum label: Arca (Senilia) senilis Linné, Lac Ahémé Dahomey, Mission Gruvel, II.1910

MNHN-IM-2022-4600

*Bucardium ringens*

Cotonou, dredging at a water depth of 20-25 meters (6.33°N, 2.39°E)

A fresh isolated valve of a juvenile.

SacA-68812: 606 ± 18 BP

$F^{14}C = 0.9273 \pm 0.0020$

$\delta^{13}C = 0.26$ ‰ VPDB

Collection date: February 1910

Collector: Mission Gruvel

Museum label: Cardium ringens, Cotonou, mer, II.1910, sac 372, Mission Gruvel

Note: The information that the sample came from a dredging at 20-25 meters water depth in front of Cotonou can be found in Dautzenberg (1912).

MNHN-IM-2022-4602

*Bucardium ringens*

La Bouche du Roi, Grand Popo, beach (6.29°N, 1.92°E)

An isolated valve with remains of the hinge ligament.

SacA-68814: 527 ± 17 BP

$F^{14}C = 0.9365 \pm 0.0020$

$\delta^{13}C = 0.25$ ‰ VPDB

Collection date: February 1910

Collector: Mission Gruvel

Museum label: Cardium ringens Gmelin, Bouche du Roi, Gd Popo, plage, II.1910, Mission Gruvel.

Note: Three labels mention the same location, but one label mentions the Catumbella estuary (Angola, June 17th 1910). We believe the sample is from Grand Popo.

**3.1.7 Samples from Ivory Coast**

MNHN-IM-2022-4595
*Bucardium ringens*
Grand Bassam, beach (5.19°N, 3.73°W)
An isolated valve with remains of the hinge ligament.
SacA-68807: 507 ± 18 BP
$F^{14}C = 0.9388 \pm 0.0021$
$\delta^{13}C = 0.29$ ‰ VPDB
Collection date: 1909-1910
Collector: Mission Gruvel
Museum label: Cardium ringens Gmelin, plage de Gd Bassam 1909-10, mission Gruvel.

MNHN-IM-2022-4597
*Bucardium ringens*
Jacqueville, beach (5.19°N, 4.42°W)
An isolated valve of a juvenile with remains of the hinge ligament.
SacA-68809: 995 ± 18 BP
$F^{14}C = 0.8835 \pm 0.0020$
$\delta^{13}C = 0.03$ ‰ VPDB
Collection date: January 10th 1910
Collector: Mission Gruvel
Museum label: Cardium ringens Gmel., Jacqueville Côte d'Ivoire, plage, 19.I.10, Mission Gruvel.

**3.1.8 Sample from Gabon**

MNHN-IM-2022-4613
*Ostrea stentina*
Port-Gentil (0.71°S, 8.79°E)
An isolated valve with remains of flesh.
SacA-68832: 497 ± 19 BP
$F^{14}C = 0.9401 \pm 0.0022$
$\delta^{13}C = 2.24$ ‰ VPDB

Collection date: 1948

Collector: Charles Roux' mission

Museum label: Port Gentil, M Roux, 1949.

Note: Charles Roux writes in 1949 (Roux, 1949) that he was in the Port-Gentil area during the year 1948. We can understand that he was already back to France in 1949. Hence, the collection date must be 1948.

### 3.1.9 Samples from Republic of Congo

MNHN-IM-2022-4614

*Ostrea stentina*

Loango (4.66°S, 11.80°E)

An isolated valve with remains of flesh.

SacA-68833: 571 ± 19 BP

$F^{14}C = 0.9314 \pm 0.0022$

$\delta^{13}C = 1.38$ ‰ VPDB

Collection date: 1890

Collector: Augusto Nobre

Museum label: Ostrea stentina Payr Loango M. Nobre 1890

MNHN-IM-2022-4604

*Donax rugosus*

Pointe-Noire (4.76°S, 11.84°E)

A fresh isolated valve from a juvenile specimen.

SacA-68816: 447 ± 18 BP

$F^{14}C = 0.9459 \pm 0.0021$

$\delta^{13}C = 0.84$ ‰ VPDB

Collection date: December 1936 – April 1937

Collector: Edgard Aubert de la Rüe

Museum label: Pte Noire, Aubert de la Rüe, 1937

Note: Edgard Aubert de la Rüe was in Congo from 18/12/1936 to 16/04/1937 as evidences by his field books kept in the archives of the Musée du Quai Branly (files 2AP/62 to 2AP/64)

### 3.1.10 Samples from Angola

MNHN-IM-2022-4590

*Senilia senilis*

Cabinda (5.55°S, 12.20°E)

Articulated specimen with well-preserved periostracum.

SacA-68822: 584 ± 17 BP

$F^{14}C = 0.9298 \pm 0.0020$

$\delta^{13}C = -1.15$ ‰ VPDB

Collection date: 1885-1887

Collector: Paul Hesse

Museum label: Cabinda, Cabinda, Angola; C.R. Boettger coll. 1909

Note: The shell was donated by Caesar R. Boettger to the MNHN in 1909 (Oliver and von Cosel, 1992) but collected earlier by Paul Hesse when Hesse was leaving in Banana (Democratic Republic of Congo) south of Cabinda (Boettger, 1912). Boettger (1912, p. 110) writes that Hesse's collection includes a number of *Senilia senilis* specimens from Cabinda. The collection date is unfortunately not provided. However, Hesse was employed by a trading company in Banana by the end/beginning 1884/1885 since at least after March 1886 (Westhoff, 1886). Mollusc specimens reported in Boettger (1912) were collected by Hesse between 1885 and 1886. Also, Hesse collected reptile specimens in Cabinda in 1885 and 1887 (Boettger, 1898). Thus, we believe that the MNHN specimen must have been collected between 1885 and 1887.

MNHN-IM-2022-4594

*Senilia senilis*

Cabinda (5.55°S, 12.20°E)

An isolated valve with well-preserved periostracum.

SacA-68826: 536 ± 19 BP

$F^{14}C = 0.9355 \pm 0.0022$

$\delta^{13}C = -3.11$ ‰ VPDB

Collection date: June 6th 1921

Collector: Unknown

Museum label [Staadt collection]: Arca senilis Lin, Cabenda, Africa, Guinea, bought just over 1d on June 6th 1921 in Grays Jun rd, (some of the specimens at B. M. are more than double the size of mine).

MNHN-IM-2022-4589

*Senilia senilis*

Luanda, beach (8.78°S, 13.27°E)

Articulated specimen with well-preserved periostracum.

SacA-68821: 538 ± 19 BP

$F^{14}C = 0.9353 \pm 0.0022$

$\delta^{13}C$ = -0.16 ‰ VPDB

Collection date: May 18th 1910

Collector: Mission Gruvel

Museum label: Arca (Senilia) senilis Linné, St Paul de Loanda, plage, Mission Gruvel, 18.V.1910.

MNHN-IM-2022-4605

*Donax rugosus*

Luanda, beach (8.82°S, 13.21°E)

Articulated specimen with remains of flesh and hinge ligament.

SacA-68817: 528 ± 18 BP

$F^{14}C$ = 0.9364 ± 0.0021

$\delta^{13}C$ = 0.82 ‰ VPDB

Collection date: May 18th 1910

Collector: Mission Gruvel

Museum label: Donax rugosus Linné, St Paul de Loanda plage, 18.V.10, Mission Gruvel.

MNHN-IM-2022-4596

*Bucardium ringens*

Bay of Lobito, near the peninsula (12.33°S, 13.56°E)

An isolated valve.

SacA-68808: 595 ± 17 BP

$F^{14}C$ = 0.9286 ± 0.0020

$\delta^{13}C$ = 1.51 ‰ VPDB

Collection date: June 1910

Collector: Mission Gruvel

Museum label: Cardium ringens Gmelin, Baie de Lobito côté presqu'île, VI.1910, mission Gruvel.

MNHN-IM-2022-4617

*Ostrea stentina*

Moçâmedes (15.18°S, 12.14°E)

A fresh isolated valve.

SacA-68836: 568 ± 18 BP

$F^{14}C$ = 0.9317 ± 0.0021

$\delta^{13}C$ = 1.75 ‰ VPDB

Collection date: 1910
Collector: Mission Gruvel
Museum label: Mossamédès, m Gruvel, 1910

### 3.2 West African marine reservoir ages

The vast majority of the calculated ΔR values, with an average of -77 ± 47 $^{14}$C yrs (1sd, n = 25), which corresponds to an averaged R value of 400 ± 59 $^{14}$C yrs (1sd, n = 25), are typical of low latitudes marine reservoir age values (Bard, 1988; Bard et al., 1994) (Table S1 in the Supplement, Fig. 1). Our results agree perfectly with those already obtained (Ndeye, 2008) from the Nouadhibou-Cansado Bay area (Mauritania; Nh in Fig. 1) and the Dakar area (Senegal; Dk in Fig. 1); the only two areas that we can compare our results to.

No significant interspecific differences were observed. This is best illustrated for the localities where reservoir age values were obtained from at least two different species for the same calendar time. In the Dakar area (Senegal; Dk in Fig. 1) for years 1908-1909, we present data for 5 species (*Bucardium ringens*, *Donax rugosus*, *Mactra glabrata*, *Ostrea stentina*, *Senilia senilis*) (Ndeye, 2008; this study) all clustering within a range of [413;546] ([min; max]) with an average ΔR value of -16 ± 56 $^{14}$C yrs (1sd, n = 6) (corresponding to an average R value of 478 ± 55 $^{14}$C yrs). This was also the case for Luanda (Angola; Lu in Fig. 1) in the year 1910, with two species (*Donax rugosus* and *Senilia senilis*) yielding the same reservoir age values. This was further supported for the area of Nouadhibou-Cansado Bay (Mauritania) showing the same pattern (Ndeye, 2008; this study), although one sample over four was likely an outlier (*Bucardium ringens* with # MNHN-IM-2022-4599). The fact that species living in very different ecological habitats (e.g., *Senilia senilis* in lagoons/semi-enclosed bays and *Donax rugosus* on beaches exposed to heavy surf; see also section material) show similar reservoir age values (R or ΔR) suggests that the habitat only exerts a minor influence on measured reservoir age. The fact that all investigated species in this study correspond to suspension feeders further implies that suspension feeders are suitable material for reservoir age reconstruction.

Unlike semi-isolated basins such as the Black Sea (Soulet et al., 2019), where the radiocarbon system is closely linked to the local carbon stable isotopic system, the open-ocean coastal region of West Africa is characterize by the lack of any relationship between reservoir age values (R or ΔR) and stable oxygen and carbon isotope compositions ($r^2$ of 0.02 and 0.001, respectively), as inferred from our results.

### 3.3 Marine reservoir evolution over time

The local marine reservoir age were averaged over 5-yrs windows ([1886-1890]-[1891-1895] and so on), excluding the five outliers discussed in section 3.5. Sample with radiocarbon lab # AA-70015 (see Table S1 in the Supplement) is a single value from 1916 and was averaged with samples from years 1912. We also calculated global marine reservoir age as the difference between the Marine20 and IntCal20 calibration curves. The evolution of the marine reservoir age of coastal West Africa (pink symbols in Fig. 2) shows a similar trend as that of the global marine reservoir age (black line in Fig. 2) with values declining steadily with time since ca. 1900 AD. The $^{14}$C age evolution of the global ocean (Marine20 calibration curve; Heaton et al.,

2020) is constructed using the global carbon cycle model BICYCLE (Köhler et al., 2006; Köhler and Fischer, 2006, 2004; Köhler et al., 2005). This box model incorporates a globally averaged atmospheric box and modules of the terrestrial (7 boxes) and oceanic (10 boxes) components of the carbon cycle. It is driven by temporal changes in the boundary conditions mimicking changing climate and simulates changes in the carbon cycle including $^{14}C$. To construct the Marine20 calibration curve, the BICYCLE model was revised to allow the atmospheric $CO_2$ and $F^{14}C$ to be specified externally (Heaton et al., 2020). The global marine reservoir age evolution calculated as the difference between Marine20 and Intcal20 calibration curves (Fig.1) is thus also an output of the revised BICYCLE model. The fact that the trend in our measured marine reservoir age is similar to that of the globally modelled ones strongly suggest that global carbon cycle and climate are the first order drivers of the coastal West African marine reservoir age rather than local effects.

### 3.4 Marine reservoir age off equatorial Ogooué and Congo rivers

Large rivers draining equatorial Africa as the Ogooué and Congo rivers inject massive amounts of freshwater into the Atlantic Ocean (Lambert et al., 2015; Milliman and Farnsworth, 2011) leading to extensive sea surface salinity negative anomalies (Martins and Stammer, 2022). The sea surface salinity negative anomalies are associated with net primary productivity positive anomalies that are likely caused by the nutrient-rich river plumes from the Ogooué and Congo Rivers (Martins and Stammer, 2022). From a radiocarbon perspective, such net primary productivity positive anomalies should imply an increased uptake of atmospheric $CO_2$ through intensified biological pump. As a result, the reservoir age should be lower than average. The Congo River represents the second largest supplier of dissolved organic carbon (DOC) to global ocean with ~5% of the land to ocean DOC flux (Spencer et al., 2016; Coynel et al., 2005; Richey et al., 2022). The DOC exported by the Congo river is $^{14}C$-modern (Marwick et al., 2015; Spencer et al., 2012) and experiments showed that 45 % of the Congo River DOC can be photo-mineralized by sunlight (Spencer et al., 2009; Richey et al., 2022). Dissolved inorganic carbon (DIC) released from the photo-mineralisation of the Congo River DOC should also be $^{14}C$-modern. Thus, this modern DOC-derived DIC should impact the marine reservoir age towards values lower compared to average. There is a lack of available data to estimate the age and flux of dissolved $CO_2$ discharged by the Congo river into the ocean (Richey et al., 2022). Nevertheless, the marine reservoir age value measured at Port-Gentil (Gabon) close to the Ogooué river outlet is lower than average value ($\Delta R = -106 \pm 63$ $^{14}C$ years, corresponding to $R = 329 \pm 21$ $^{14}C$ yrs) (PG in Fig. 1). The marine reservoir age measured in Pointe-Noire (Republic of Congo) ~150 km north of the Congo river outlet is also lower than average ($\Delta R = -156 \pm 64$ $^{14}C$ yrs; $R = 289 \pm 20$ $^{14}C$ yrs) (PN in Fig. 1). These values could be interpreted as having been influenced by the Ogooué and Congo Rivers discharges. However, all other localities close to the Congo River outlet had marine reservoir age close to the average (Lo, Ca and Lu, in Fig. 1). Instead the lower values observed in Port-Gentil (Gabon) and Pointe-Noire (Republic of Congo) are from years 1948 and 1937 suggesting that these lower values are in line with the declining global marine reservoir evolution linked to global climate and carbon cycle changes (see section 3.3). The impact of the African equatorial rivers on the local/regional coastal marine reservoir age, if any, cannot be inferred from our results.

### 3.5 Outlier specimens

Mean marine reservoir age values (R and ΔR) are provided for West Africa based on our data, excluding 5 samples. These particular samples display much larger values with ΔR values ranging from 209 to 454 $^{14}$C yrs or R values ranging from 701 to 912 $^{14}$C yrs. Three specimens out of the five outlier samples correspond to *Bucardium ringens* specimens. We analysed 8 *Bucardium ringens* specimens. These 3 outlier specimens display reservoir age (R and ΔR) values that clearly disagree with neighbouring data (Nouadhibou-Cansado Bay, Loos Islands and Ivory Coast areas; Nh, LI and IC in Fig. 1). The Museum number of these specimens are MNHN-IM-2022-4597, MNHN-IM-2022-4599 and MNHN-IM-2022-4601. We do not expect that these larger values compared to those for neighbouring individuals come from the species feeding practice as they are all suspension feeders like all other investigated specimens. Similarly, we showed that the difference in the habitat does not impact the specie reservoir ages. Instead *Bucardium ringens* lives in the open coast from 5-10 meters to about 50 meters depth. Shells are commonly cast ashore on beaches but live-taken specimens are rare (von Cosel and Gofas, 2019). Two of the samples that are outlier were collected at low tide (Roume Island in the Loos Islands; Republic of Guinea) or on a beach (Jacqueville; Ivory Coast). It is thus possible that these outlier samples were transported subfossil samples that died a century or more before collection date. Nevertheless, we cannot fully rule out that these higher values represent some variability in the local marine reservoir age. Although, five *Bucardium ringens* samples out of eight displayed reservoir age values in agreement with the neighbouring reservoir age values, this specie might not be the best suited for reservoir age reconstruction or for sediment/archaeologic dating.

The two remaining outliers are *Ostrea stentina* specimens from the El Jadida area (Morocco; eJ in Fig. 1). The sample from El Jadida beach was a single valve looking fresh and collected from the beach (museum # MNHN-IM-2022-4609). Based on the older $^{14}$C age of this specimen, we cannot rule out that this sample could actually be a subfossil specimen. The specimen with museum # MNHN-IM-2022-4608 collected in the Sidi Moussa lagoon (south of El Jadida) was a specimen with the articulated valves and remains of flesh was still inside the shell, meaning the specimen was still alive when collected. Variations in the reservoir age could be explained by coastal upwelling that impacts some regions of the Atlantic coast of Morocco and Western Sahara (Freudenthal et al., 2001; Barton et al., 1998). Upwelled waters can be depleted in $^{14}$C relative to the sea surface potentially causing larger reservoir age values (R or ΔR) like off Portugal (Monge Soares, 1993; Monge Soares and Alveirinho Dias, 2006), California (Kennett et al., 1997), or Southern Arabian coast (Southon et al., 2002). Conversely, upwelled waters can also be nutrient-rich causing intensified ocean $CO_2$ uptake through enhanced primary production and biological pump (Williams and Follows, 2011), in that case, one could expect low-latitude average or decreased reservoir age values (R or ΔR). Off Morocco and Western Sahara, the second hypothesis appears most likely as coastal upwelling in this area is known to bring nutrient-rich waters to the surface ocean (Barton et al., 1998; Freudenthal et al., 2001), although to our knowledge no direct measurement of the $^{14}$C content of coastal waters in this region has been published yet. However, according to recent studies the El Jadida area is only weakly impacted by upwelling (Lourenço et al., 2020; Cropper et al., 2014), suggesting average reservoir age values instead of larger ones. Another explanation could be linked to the local

hydrology of the Sidi Moussa lagoon. Despite the lagoon is permanently connected to the ocean, it receives waters from rainfall and resurgences that can have an impact on the salinity in the upstream section of the lagoon (Cheggour et al., 2001). As the surrounding rocks are calcareous sandstones (Manaan, 2003), one could hypothesise that freshwaters feeding the lagoon might be depleted in $^{14}C$ due to carbonate dissolution in the lagoon watershed causing a hardwater effect and thus a larger reservoir age. A last explanation could be due to an imperfect cleaning of the shell. For *Ostrea stentina*, sediment can be trapped between the growing layers of the shell. If this sediment contains old detrital carbonates and was not perfectly removed before $^{14}C$ measurement, the $^{14}C$ age of the shell will appear older, and the reservoir age larger. Additional reservoir age reconstructions from this region on different species would be require to validate the larger reservoir age values reconstructed from the El Jadida area.

## 4 Conclusion

The analysis of pre-bomb suspension-feeding bivalves collected along coastal West Africa from 33°N to 15°S provides marine reservoir ages that are quite homogenous, with a mean ΔR value of -77 ± 47 $^{14}C$ yrs (1sd, n = 25) and a mean R value of 400 ± 59 $^{14}C$ yrs (1sd, n = 25). When including the robust dataset from Ndeye (2008), the resulting mean ΔR and R values for coastal West Africa are -62 ± 51 $^{14}C$ years (1sd, n = 33) and 416 ± 61 $^{14}C$ years (1sd, n = 33), respectively. We show that the marine reservoir age of coastal West Africa is mainly driven by the global carbon cycle and climate rather that by local effects. Our results for different species yield similar marine reservoir age values, indicating that the ecological habitat only has a second-order impact on the reservoir age reconstruction, if any. Nevertheless, we suspect that *Bucardium ringens* might not be best suited for marine reservoir age reconstruction as corresponding shells are typically not found alive on sample collecting sites. Additionally, ages obtained on *Ostrea stentina* could be possibly influenced by the presence of sediment within the growing shell layers if not fully removed after the cleaning process.

Despite these new data, large portions of the West African coast still remain to be investigated for reservoir age reconstructions, in particular off Western Sahara and Canarias Islands, Sierra Leone-Liberia, Nigeria and Namibia.

## Author contribution statement

GSoulet designed the study and raised the funding. SG, PM, GSoulet and GSiani selected the specimens in the MNHN collections. GSoulet carried out the sample preparation with assistance of ML and FF. FD performed stable isotopes measurements. GSoulet performed reservoir calculations and analysed and discussed the data with GB, GSiani and SG. GSoulet wrote the manuscript with inputs from all co-authors.

**Acknowledgements**

We are grateful to the Muséum National d'Histoire Naturelle (Paris, France) for providing the samples. We thank the LMC14 staff (Laboratoire de Mesure du Carbone-14) and the CNRS-INSU ARTEMIS national radiocarbon AMS facility for providing radiocarbon measurements published in this study. The study was funded by the French national program INSU-LEFE (ResWA project – Pre-bomb Reservoir ages for the Western coast of Africa). I (GS) thank Ifremer in Brest and Sète for supporting my research. Finally, this study was carried out and the paper written during my part-time parental leave. I dedicate this article to Marian, with whom I have had a wonderful time during this year, growing up together, him as a toddler and me as a father.

**Competing interests**

The authors declare that they have no conflict of interest.

**Data availability**

All data present in the paper are available in the text section 3.1 and in Table S1 in the Supplement.

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

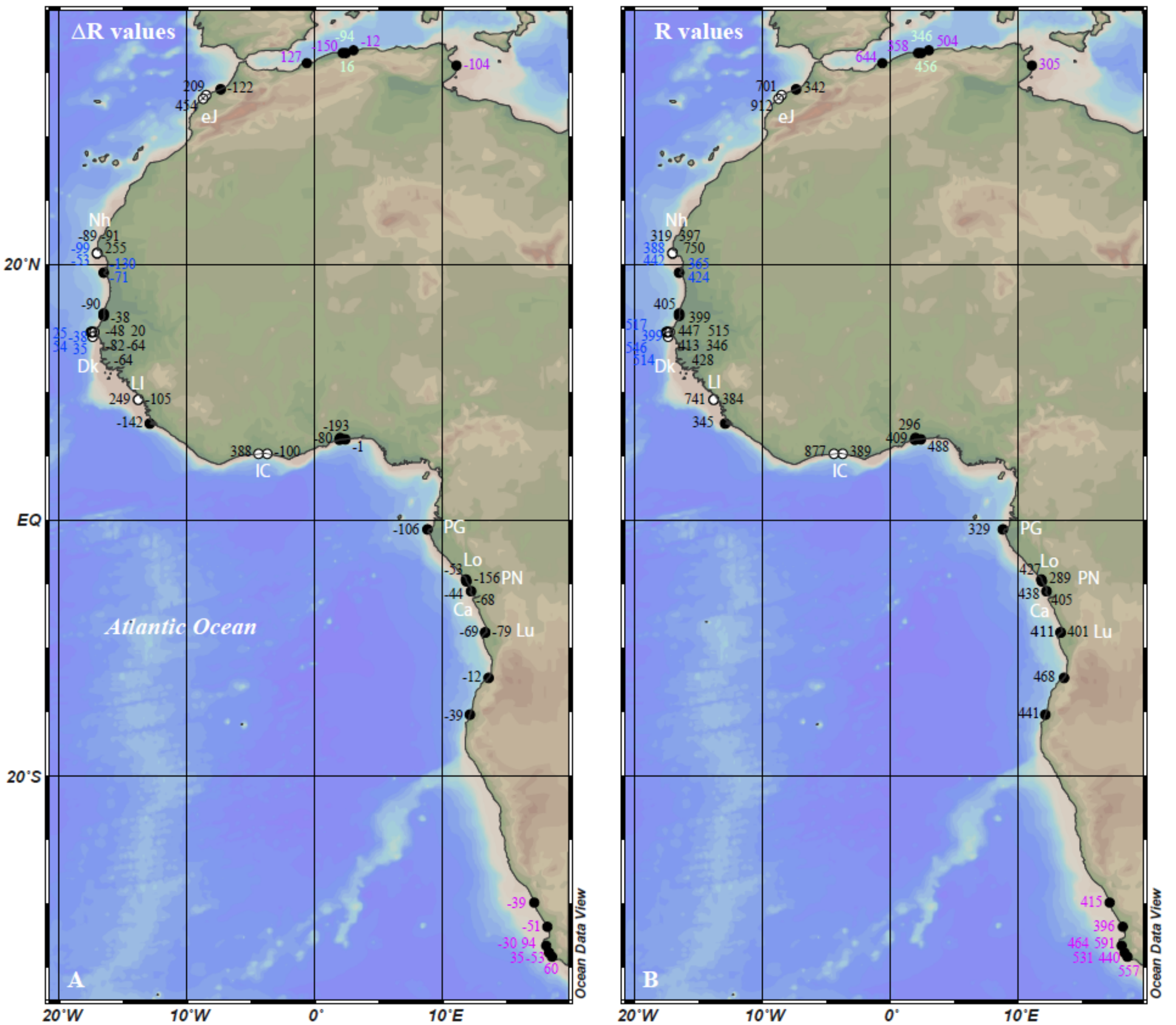

**Figure 1: The geographic distribution of marine reservoir age values along the West African coast. A. ΔR values. B. R values. Data shown in black are from this study. Other are selected results from previous studies discussed in the text, converted from their original format (conventional 14C ages and collection dates) to ΔR and R values using the latest calibration curves Marine20 (Heaton et al., 2020) and Intcal20 or SHcal20 (Reimer et al., 2020; Hogg et al., 2020), respectively. Data in blue are from Ndeye (2008), data in green are from (Reimer and McCormac, 2002), data in purple are from (Siani et al., 2000) and data in pink are from (Dewar et al., 2012). eJ, Nh, Dk, LI, IC and Lu stand for el Jadida (Morocco), Nouadhibou (Mauritania), Dakar (Senegal), Loos Islands (Republic of Guinea), Ivory Coast and Luanda (Angola). The map was drawn using Ocean Data View (Schlitzer, Reiner, Ocean Data View, https://odv.awi.de, 2022).**

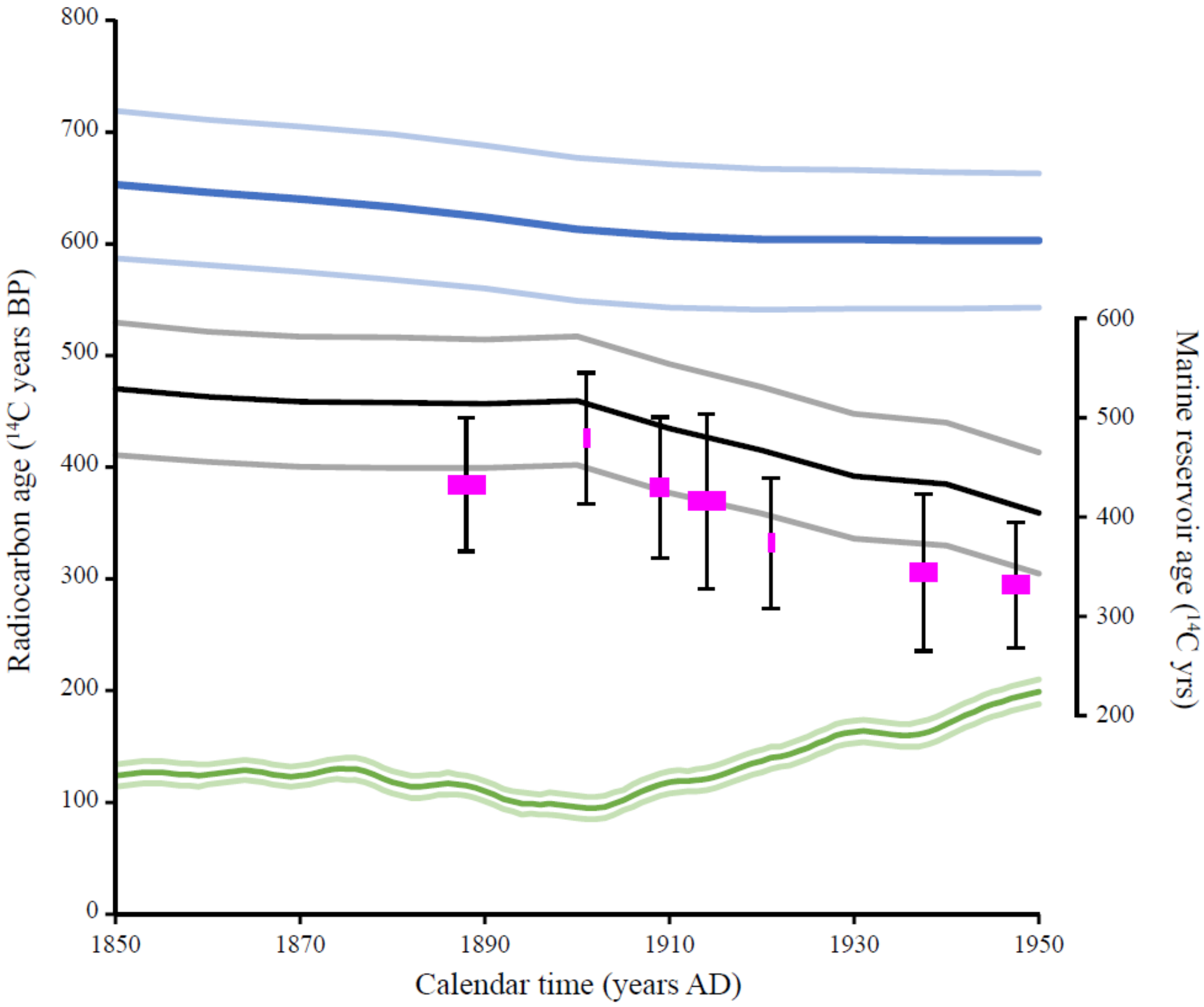

820 **Figure 2: Left axis: The radiocarbon age evolution of the atmosphere (IntCal20; green curve with its light green 1-σ envelope) and of the global ocean (Marine20; blue curve with its light blue 1-σ envelope) between 1850 and 1950 AD. Right axis: The global marine reservoir age (black curve with its grey 1-σ envelope) calculated as the difference between Marine20 and Intcal20 curves. Pink symbols are the coastal West African marine reservoir age calculated averaging data over 5-yrs windows. The reported error bars are the maximum of the standard deviation of the averaged data and the individual uncertainty of the averaged data.**