# Peer review of "Marine reservoir ages for coastal West Africa"

_Geochronology, 2023_

## Referee Comment (RC1)

**Review on**

*Marine reservoir ages for coastal West Africa*

**by G. Soulet et al.**

submitted to *Geochronology*,

Submission ID 10.5194/gchron-2023-5

**Date: April 12, 2023**

In this paper 30 new $^{14}$C ages of pre-bomb bivalues from coastal West Africa with known ages between 1850 and 1950 AD are presented and discussed. The case is made that since the derived marine reservoir ages (MRA) follow Marine20 (which itself is a carbon cycle based interpretation of IntCal20) that carbon cycle and climate are responsible for the observed trend and not local effects.

While I find the data of interest and certainly worth publishing I disagree with the final conclusion. Furthermore, I believe some careful revision is necessary to explain certain details of the draft more closely in order to make the work repeatable.

1. **Trend in MRA:** Stated in the abstract and conclusions, and more widely in section 3.3 it is said that the trend in measured MRA is similar to the modelled global trend in Marine20, and since this is based on simulations with a carbon cycle model, the trend in the new data should according to the authors also be based on carbon cycle change. Unfortunatelxy, this is not the case. The trend in MRA in Marine20 between 1900 and 1950 AD is soley based on the decrease in IntCal20 (atmospheric $\Delta^{14}$C), while Marine20 (global surface ocean $\Delta^{14}$C) is constant. This is also seen in Figure 2. Furthermore, if one goes to details of the paper describing Marine20 (Heaton et al., 2020), Figure 7 contains model results in which $CO_2$ and climate are kept constant. In these runs the calculatd MRA changes similarly than in the full carbon cycle setup. This is not easily visible in this Figure 7b of the Marine20 paper, but one can check it by downloading the corresponding data from PANGAEA following the data link given in Heaton et al. (2020). I include a figure of these MRA in Marine20 (zoom-in of Figure 7b in (Heaton et al., 2020)) below. Thus, the sole

reason for the observed trend in MRA in West Africa seems to me to be the change in atmospheric $\Delta^{14}C$, which is also a global, and not a local, effect .

2. **Symbols and units:** I find it rather confusing that the authers choose to label the radiocarbon age with $^{14}C$, whose units would be $^{14}C$ years. Normally (in physics), $^{14}C$ is the amount of radiocarbon with units "number of atoms" or "number of mol". Thus, I suggest to change this labelling. However, maybe this is also a community issues (data vs model), but it might help if the same symbols are used as in other papers, check, for example the symbols in the Marine20 paper (Heaton et al., 2020) or its recently published discussion on "how-to-use-Marine20" (Heaton et al., 2022). Also, if *time* is addressed it should always be stated if "$^{14}C$ years" or "calender years" are meant, and using only "years" should be avoided in such a paper. One example, where this is missing is Table S1 in the SI, column M showing $^{14}C$ age, units should be "$^{14}C$ yrs BP".

3. **Radiocarbon results (section 3.1):** I am not familar with data reporting, maybe this detailed description is common, but my feeling is, this section is just a long version of Table S1. Indeed, some information of the text is missing in the Table and I suggest to include them there (museums label, collector). However, I have the feeling it would serve the paper better, if a condensed version of the Table appears in the main text instead of the long description and an extended version is still published as SI. But as I said, I am no expert here, so do as common in the community and ignore this comment if you feel it is rather strange. If you keep the text, however, some changes are necessary: (a) the true measured value is $F^{14}C$, not $^{14}C$ age. So I believe, that $F^{14}C$ should be mentioned directly after the "radiocarbon laboratory number"; (b) the $^{14}C$ age now appearing after the "radiocarbon laboratory number" comes without label of what it is and the units are wrong (units are "BP", and should be "$^{14}C$ yrs BP").

4. **Calculated mean values:** At the beginning of section 3.2. it is not clear which 25 samples are averaged, since there should be 30 new samples and the SI table contains 38 samples. I believe what was done is averaging only the new (own)

data without the outliers. However, this is not said so. Outlieres are discussed later, so I suggest to bring outlieres first and only thereafter make average values without them. The outlieres are also not marked in the SI table, so it is not possible for me to reproduce the stated averaging without a lot of digging in the relevant section on outliers. Furthermore, you average samples with errors, for which to my knowledge a weighted mean is best used as done also in calculations of mean values from the marine radiocarbon reservoir database (http://calib.org/marine/) See http://calib.org/marine/AverageDeltaR.html for details on errors. Even when weighted means are not taken (for which the reader might then want to be given an argument for this omission) it needs to state clearly on what the calculated error is based on. Is this only the error from the averaging or the mean error of the individual errors?

5. **Figure 2:** Here radiocarbon age (left y-axis for IntCal20 (green) and Marine20 (blue)) and MRA (right y-axis for Marine20 (black) and magenta data points) are mixed. I strongly suggest to split the figure in two to make it easier for the reader to see which axis needs to be used for which data sets.

**References**

Heaton, T. J., Köhler, P., Butzin, M., Bard, E., Reimer, R. W., Austin, W. E. N., Ramsey, C. B., Grootes, P. M., Hughen, K. A., Kromer, B., Reimer, P. J., Adkins, J., Burke, A., Cook, M. S., Olsen, J., and Skinner, L. C.: Marine20 — the marine radiocarbon age calibration curve (0–55,000 cal BP), Radiocarbon, 62, 779–820, doi: 10.1017/RDC.2020.68, 2020.

Heaton, T. J., Bard, E., Bronk Ramsey, C., Butzin, M., Hatté, C., Hughen, K. A., Köhler, P., and Reimer, P. J.: A response to community questions on the Marine20 radiocarbon age calibration curve: marine reservoir ages and the calibration of 14C samples from the oceans, Radiocarbon, pp. 1–27, doi:10.1017/RDC.2022.66, 2022.

[Figure]

Figure 1: Zoom-in on Figure 7b of Heaton et al. (2020).

---

## Author Comment (AC1)

Response to Paula Reimer's Community Comments

The presentation of marine reservoir ages for the west coast of Africa by Soulet et al. provides much needed data for calibrating radiocarbon ages of carbonates from marine organisms for this region. While the main results are not unexpected, they will give confidence to archaeologists and geoscientists using radiocarbon dates of shells or foraminifera for age models. It is interesting that two shells from near the discharge of the nutrient-rich Ogooué and Congo Rivers had lower reservoir ages than nearby shells but having been collected in the 1930s and 1940s fit the global decline in R. It would be worth mentioning that this global decline is due to fossil fuel input to the atmosphere (e.g. Druffel & Suess 1983). The manuscript is well-written and contains important details on the samples used that are often omitted in publications. I would definitely recommend publishing this paper with minor corrections.

We thank Paula Reimer for her positive comments.

  i)   Indeed, the main results are not surprising but are useful for the $^{14}$C dating community;
  ii)  We will mention that the most recent R values are in line with the global R decline due to fossil fuel input to the atmosphere. This comment is actually related to one of the major comments of Anonymous Reviewer. It will be fully addressed in our response to her/him. The revised manuscript has been modified accordingly (Please, see L22, L534-538, L565);
  iii) We thank the reviewer for acknowledging that the information regarding the studied shells is important while often omitted in publications.

Below we address Paula Reimer's specific comments. "LXX" refers to the lines in the revised manuscript with tracks. Note that the version of the revised manuscript will be checked for English by a native speaker.

Specific comments.

Line 41: 'Larger ΔR values are located at high-latitudes'. I would suggest qualifying this as 'Most larger ΔR values' as there are low-latitude locations where ΔR values are higher due to groundwater carbonates being leached into coastal water (e.g. Hadden & Cherkinsky 2015) or where upwelling increases the ΔR values (e.g. Gulf of California, Goodfriend & Flessa 1997).

Thanks for comment and references. We modified the text accordingly. Please, see L46 in the revised manuscript (with tracks version).

Line 501-504: 'The fact that species living in very different ecological habitats (e.g., Senilia senilis in lagoons/semi-enclosed bays and Donax rugosus on beaches exposed to heavy surf; see also section material) show similar reservoir age values (R or ΔR) suggests that the habitat only exerts a minor influence on measured reservoir age.' There are many examples where habitat does exert a major influence on R so please clarify that this conclusion is for the regions studied.

We modified the manuscript to clarify this point. We now clearly state that for most species presented here, the habitat seems to exert only a minor influence for the regions studied. Please, see the revised version of our manuscript: L19, L515, L575 and L622.

Line 561-562: 'It is thus possible that these outlier samples were transported subfossil samples that died a century or more before collection date'. Two of these outlier samples were listed as having ligaments attached so it seems very unlikely that they had been dead for very long before collection.

Right. Two of these samples had very small remains of the hinge ligament. It may be possible that the hinge ligament could be preserved in case of very favourable environmental conditions (Forman et al., 2004; Huntley et al. 2021). Alternatively, it may be also possible that unlike the other studied species here, the habitat exerts an influence on R and ΔR values measured in *B. ringens*. We balanced the discussion regarding *B. cardium* in order to mention this possibility. Please, see L580-586.

Forman et al 2004 QSR

Figure 1 caption: please specify what PG, LO, PN and Ca stand for on the map.

Indeed, we forgot to specify the information in Figure 1 caption. We modified this accordingly in the revised version. Here, just for information: Port-Gentil (PG), Loango (Lo), Pointe-Noire (PN), Cabinda (Ca).

See, L867-869 in the revised manuscript with tracks.

There are also some grammatical errors that need to be corrected by a fluent English speaker. Also the word 'specie' is not correct as 'species' is both singular and plural.

We thank Paula Reimer who emailed us an annotated pdf file of our original submission with the grammatical and typos she spotted in. We modified the manuscript accordingly. The revised version of our manuscript being substantially amended, it will be checked by a native English speaker before resubmission.

DRUFFEL, E. M. & SUESS, H. E. 1983. On the Radiocarbon Record in Banded Corals - Exchange Parameters and Net Transport of (Co2)-C-14 between Atmosphere and Surface Ocean. *Journal of Geophysical Research-Oceans and Atmospheres,* 88**,** 1271-1280.

GOODFRIEND, G. A. & FLESSA, K. W. 1997. Radiocarbon reservoir ages in the Gulf of California: Roles of upwelling and flow from the Colorado River. *Radiocarbon,* 39**,** 139-148.

HADDEN, C. S. & CHERKINSKY, A. 2015. 14C variations in pre-bomb nearshore habitats of the Florida Panhandle, USA. *Radiocarbon,* 57**,** 469-479.

---

## Author Comment (AC2)

**Review on**

*Marine reservoir ages for coastal West Africa* **by G. Soulet
et al.**

submitted to *Geochronology*,

Submission ID 10.5194/gchron-2023-5

**Date: April 12, 2023**

In this paper 30 new [14]C ages of pre-bomb bivalues from coastal West Africa with known ages between 1850 and 1950 AD are presented and discussed. The case is made that since the derived marine reservoir ages (MRA) follow Marine20 (which itself is a carbon cycle based interpretation of IntCal20) that carbon cycle and climate are responsible for the observed trend and not local effects.

While I find the data of interest and certainly worth publishing I disagree with the final conclusion. Furthermore, I believe some careful revision is necessary to explain certain details of the draft more closely in order to make the work repeatable.

We thank Anonymous Reviewer for her/his comments and especially regarding the first comment that corrects our section 3.3. Below is our response to each comment.

Note that "LXX" refers to lines in the revised manuscript with tracks. The revised manuscript will be checked for English by a native speaker.

1. **Trend in MRA:** Stated in the abstract and conclusions, and more widely in section 3.3 it is said that the trend in measured MRA is similar to the modelled global trend in Marine20, and since this is based on simulations with a carbon cycle model, the trend in the new data should according to the authors also be based on carbon cycle change. Unfortunately, this is not the case. The trend in MRA in Marine20 between 1900 and 1950 AD is solely based on the decrease in IntCal20 (atmospheric $\Delta^{14}$C), while Marine20 (global surface ocean $\Delta^{14}$C) is constant. This is also seen in Figure 2. Furthermore, if one goes to details of the paper describing Marine20 (Heaton et al., 2020), Figure 7 contains model results in which $CO_2$ and climate are kept constant. In these runs the calculated MRA changes similarly than in the full carbon cycle setup. This is not easily visible in this Figure 7b of the Marine20 paper, but one can check it by downloading the corresponding data from PANGAEA following the data link given in Heaton et al. (2020). I include a figure of these MRA in Marine20 (zoom-in of Figure 7b in (Heaton et al., 2020)) below. Thus, the sole reason for the observed trend in MRA in West Africa

seems to me to be the change in atmospheric $\Delta^{14}C$, which is also a global, and not a local, effect.

Many thanks for raising the point with a crystal-clear explanation. This is also related to one of Paula Reimer's comments. We will amend the main text and abstract accordingly to state that the observed trend in the MRA in West Africa is related to the change in the atmospheric $\Delta^{14}C$ linked to fossil fuel burning. Please, see L22, L534-538, L565.

2. **Symbols and units:** I find it rather confusing that the authors choose to label the radiocarbon age with $^{14}C$, whose units would be $^{14}C$ years. Normally (in physics), $^{14}C$ is the amount of radiocarbon with units "number of atoms" or "number of mol". Thus, I suggest to change this labelling. However, maybe this is also a community issues (data vs model), but it might help if the same symbols are used as in other papers, check, for example the symbols in the Marine20 paper (Heaton et al., 2020) or its recently published discussion on "how-to-use-Marine20" (Heaton et al., 2022). Also, if *time* is addressed it should always be stated if "$^{14}C$ years" or "calender years" are meant, and using only "years" should be avoided in such a paper. One example, where this is missing is Table S1 in the SI, column M showing $^{14}C$ age, units should be "$^{14}C$ yrs BP".

We followed the radiocarbon community conventions to report the radiocarbon ages ("BP", or "$^{14}C$ yrs BP") and the reservoir ages ("$^{14}C$ yrs"). Regarding the collection dates, it seemed very obvious to us that they are calendar ages, because they are related to the day/year when a physical person collected the sample: see https://www.cambridge.org/core/journals/radiocarbon/information/author-instructions/preparing-your-materials

Modified accordingly the unit in column M of Table S1.

3. **Radiocarbon results (section 3.1):** I am not familar with data reporting, maybe this detailed description is common, but my feeling is, this section is just a long version of Table S1. Indeed, some information of the text is missing in the Table and I suggest to include them there (museums label, collector). However, I have the feeling it would serve the paper better, if a condensed version of the Table appears in the main text instead of the long description and an extended version is still published as SI. But as I said, I am no expert here, so do as common in the community and ignore this comment if you feel it is rather strange.

Precisely because the information regarding the sample provenance and condition is generally omitted in publications, we have decided to provide as much as information possible in the main text of our paper. Moreover, we want that this long description including the numerical results appears in the main text, not in the supplement, because the information that appears in this section (3.1) is the most important. It is actually the

only one that is needed to repeat our work and recalculate the reservoir age values when the next updates of the calibration curves will be released.

If you keep the text, however, some changes are necessary: (a) the true measured value is $F^{14}C$, not $^{14}C$ age. So I believe, that $F^{14}C$ should be mentioned directly after the "radiocarbon laboratory number"; (b) the $^{14}C$ age now appearing after the "radiocarbon laboratory number" comes without label of what it is and the units are wrong (units are "BP", and should be "$^{14}C$ yrs BP").

We understand the above comments, however we did follow the conventions to report $^{14}C$ dates as recommended in the journal *Radiocarbon*:

https://www.cambridge.org/core/journals/radiocarbon/information/author-instructions/preparing-your-materials

The radiocarbon lab ID of the sample is followed by the $^{14}C$ date with the unit BP. Only the calibrated dates need "cal yrs BP", calendar dates only need "AD" (in our case). We added the information (see e.g., L151, L398…)

**Calculated mean values:** At the beginning of section 3.2. it is not clear which 25 samples are averaged, since there should be 30 new samples and the SI table contains 38 samples. I believe what was done is averaging only the new (own) data without the outliers. However, this is not said so.

Right, in section 3.2, we averaged only our own set of data excluding the five outliers. In the revised manuscript, we will make the point clear in both the text and Table S1. The outliers are now clearly flagged in Table S1.

Outlieres are discussed later, so I suggest to bring outlieres first and only thereafter make average values without them.

We feel more appropriate to discuss first the non-outlier data before the outliers because an outlier can only be detected from a larger set of data. We hope our decision is acceptable for the Anonymous Reviewer.

The outlieres are also not marked in the SI table, so it is not possible for me to reproduce the stated averaging without a lot of digging in the relevant section on outliers.

As stated above, we will make it easy for the reader to recalculate our averaging, clearly indicating the methodology and flagging outliers in Table S1.

Furthermore, you average samples with errors, for which to my knowledge a weighted mean is best used as done also in calculations of mean values from the marine radiocarbon reservoir database (http://calib.org/marine/)

See http://calib.org/marine/AverageDeltaR.html for details on errors. Even when weighted means are not taken (for which the reader might then want to be given an argument for this omission) it needs to state clearly on what the calculated error is based on. Is this only the error from the averaging or the mean error of the individual errors?

We thought the methodology was clear enough as we wrote, e.g. "an average of -77 ± 47 $^{14}$C yrs 1sd, n = 25)," which means we took the averaged value of 25 individual values ("n = 25"), and that the reported error is the standard deviation of the averaged values ("1sd"). Nevertheless, we will alter the text to make the averaging methodology crystal clear, and comply to the recommended methodology described in the marine radiocarbon database: i) take the weighted mean by variance, ii) the reported uncertainty is the maximum of the Standard Deviation of ΔR values and the weighted uncertainty in mean of ΔR values. Please, see L500-502.

4. **Figure 2:** Here radiocarbon age (left y-axis for IntCal20 (green) and Marine20 (blue)) and MRA (right y-axis for Marine20 (black) and magenta data points) are mixed. I strongly suggest to split the figure in two to make it easier for the reader to see which axis needs to be used for which data sets.

Thanks. We modified our Figure 2 accordingly.

**References**

Heaton, T. J., Ko¨hler, P., Butzin, M., Bard, E., Reimer, R. W., Austin, W. E. N., Ramsey, C. B., Grootes, P. M., Hughen, K. A., Kromer, B., Reimer, P. J., Adkins, J., Burke, A., Cook, M. S., Olsen, J., and Skinner, L. C.: Marine20 — the marine radiocarbon age calibration curve (0–55,000 cal BP), Radiocarbon, 62, 779–820, doi: 10.1017/RDC.2020.68, 2020.

Heaton, T. J., Bard, E., Bronk Ramsey, C., Butzin, M., Hatt´e, C., Hughen, K. A., Ko¨hler, P., and Reimer, P. J.: A response to community questions on the Marine20 radiocarbon age calibration curve: marine reservoir ages and the calibration of 14C samples from the oceans, Radiocarbon, pp. 1–27, doi:10.1017/RDC.2022.66, 2022.

[Figure]

Figure 1: Zoom-in on Figure 7b of Heaton et al. (2020).

---

## Author Comment (AC3)

gchron-2023-5 Marine reservoir ages for coastal West Africa by Soulet & al.

The article by Soulet & al presents the 14C reservoir age measurements of seawater for the western coast of Africa, from Morocco to Angola. This type of data is lacking in the community and therefore deserves to be published. The calculation of reservoir ages and ΔR does not require any additional comment. Nevertheless, some points remain obscure and therefore call for further clarification.

We would like to thank Michel Fontugne for his constructive comments. We reply in details to each of them below. "LXX" refers to the lines in the revised manuscript with tracks.

Sampling.

As for the results published by Ndeye (2008), a large number of samples come from the MNHN Paris collections and more particularly from the Gruvel missions. Soulet & al. classify five results as aberrant, in particular those from Guinea and the Ivory Coast which were collected by dredging (see Dautzenberg, Annales Inst. Océanogr.). This method of collection is not the most appropriate for this kind of study….

We agree that dredging is not the best appropriate way to collect live samples for reservoir age reconstruction. However, we have to deal with this limitation as we rely on museum samples older than AD 1950.

However, here specifically, not all samples from Guinea and Ivory Coast were collected through dredging. Please, have a careful check at the reference you cite. We also used and cited this reference to check the provenance and collection method of each sample from the Gruvel mission. Actually, only one sample out of 30 used here was collected through dredging and it was from Cotonou (Benin) (sample MNHN-IM-2022-4600), and it is not an outlier. The information was already mentioned in our original submission.

Ndeye had obtained 6 aberrant results including 4 for samples from the MNHN.

Well, that is not a strong evidence to disqualify samples from the MHNH collections. Instead, we would like to emphasize that the reservoir age reconstructions for the Pacific Ocean (Southon et al., 2002), for the French Coast (Tisnérat-Laborde et al., 2010), for the Black Sea and Mediterranean Coasts (Siani et al., 2000), from Kerguelen Islands (Paterne et al., 2019) are from the MNHN and yielded good results despite there were certainly some outliers.

The sample from Jacqueville and Ile de Roume come from the shell of B. ringens, which could encourage us to requalify this species for this type of study.

Owing to P Reimer' comments and yours, we balanced our discussion about *B. ringens*. Thanks. Please, see L23-24 and L580-586.

Outlying samples.

The other so-called aberrant samples are collected on coasts subject to the influence of deep water rises (upwelling) whose CID is depleted in 14C. The values obtained from Morocco to

Dakar are compatible or characteristic of these upwelling zones. For comparison, the authors can refer to the reservoir ages calculated for the Peruvian upwelling (Kennett & al, 2002; Fontugne & al, 2004; Jones & al 2007, 2010; Owen, 2002, in Radiocarbon Ortlieb & al QR 2011; Etayo -Cadavid et al Geology 2013). Surprisingly, these references are absent when the authors mention the zones of deep water upwelling (see in Outlier specimens). The works cited above demonstrate the extreme variability of the values of the reservoir ages depending on the position of the upwelling cells, the intensity of the winds but also a variability during the period of life of the mollusk.

Indeed, we did not cite these papers and we will do so in the revised manuscript. Thanks. Please, see L597.

The sample from Lake Ahémé (Benin) with a $\partial$13C of -4.76‰ seems to have been marked by a strong contribution of carbon of continental origin likely to reduce the reservoir age value. As this lake is located 10km from the coast with narrow communication with the coastal lagoons, it cannot really be considered as representative of the Atlantic Ocean. It would be better not to take it into account but the result can be published in the table S1.

Right, we will remove this sample from the regional averaged value, and write a note in the sample description stating why. Please, see L335-337.

This sample comes from the Gruvel mission. Given the number of aberrant samples provided by this mission to the authors as well as to Ndeye (2008), it would be desirable to verify the harvesting conditions for all these samples.

Ndeye (2008) reported 6 outliers. However, only one was from the Gruvel mission.

We have already checked each provenance. We had 5 outliers out of 30 samples, among which two could have been influenced by upwelling conditions. So, we had only 3 outliers out of 30 samples. Right, they are all from the Gruvel mission, but the vast majority of our samples comes from the Gruvel mission… But, what is important is to flag and discuss the potential outliers. We feel it is exactly what we did in this manuscript.

References.

Not all bibliographic references are adequate: Stuiver et al 1986, Stuiver & Brasiunas 1993 (both in Radiocarbon) would be more justified for the definitions of R and ΔR (as mentioned later in the text). Stuiver and Pollack (1977) only specify that reservoir age corrections should not enter into the calculation of conventional 14C dates.

Thanks, we will modify the main text accordingly. Please, see L28-29.

As with ocean/atmosphere $CO_2$ exchanges, it would be more accurate to mention the pioneering works of Revelle & Suess, 1957 and Craig, 1957 (both in Tellus). Brocker & al., 1985 and Stuiver 1980 (both in JGR) could also have been mentioned; these studies giving the distribution of 14C in the ocean and highlight the equatorial upwelling (or divergence) characterized by low 14C water. Of course, Bard et al 1988 can be added.

Fair. We will modify the main text so that these references are cited. Please, see L34-35.

The reference to the Black Sea, which is a lake occasionally connected to the Mediterranean Sea, is not the most judicious. A reference to the Baltic Sea (see Lougheed & al Clim. Past 2013) would reinforce this point. These remarks are not exhaustive, I think that the authors will be able to complete the bibliographical references with more precision.

We added the Baltic Sea reference to the manuscript. Please, see L518.

Stable Carbon Isotopes and reservoir ages uncertainties.

The authors indicate reproducibilities of ±0.04‰ and ±0.02‰ for the values of δ18O and δ13C, respectively. These are occasional measurements integrating a seasonal variability of 0.5 to 1‰ recorded by the growth rings of the shell (see among others Carré & al, 3P 2005, Jones & al RC, 2007, 2010.

Right, we do indicate that this is only the instrument reproducibility. Although our stable isotope measurements are not intended to discuss any seasonal variability, we will add a sentence in the revised manuscript indicating that seasonal environmental variability can impact the stable isotopic value over a range larger than the instrument reproducibility, citing the relevant literature. Please, see L120-122.

Jones' work also shows a seasonal variation of about 100 years in marine reservoir age. Perhaps, the authors could comment on the evolution of reservoir ages between 1890 & 1950 by reconsidering this new level of uncertainty.

Similarly, we can add a sentence indicating the extent of variability observed in Jones' work. However, we do not have clear results to firmly state this, and it could be seen as an over-interpretation of the results. Regarding the evolution between 1890 and 1950, the averaged values have already uncertainties larger than 100 $^{14}$C yrs, still showing a clear decreasing trend in the reservoir age.

We still used these results in the section outliers: Please, see L580-586.

In detail Table S1. Columns F and G should simply be labeled Latitude and Longitude, the minus sign indicating S and W respectively.

Modified accordingly.

The authors wrote "*Upwelled waters can be depleted in 14C relative to the sea surface*" ; but upwelled water are always depleted in 14C. The upwellings areas are the atmosphere's largest natural source of CO2 (see among others Takahashi & al Deep Sea Res., 2002).

We reworded the sentence accordingly. Please, see L595.

To conclude, this paper needs more precisions. Nevertheless, its data are needed.

Thanks, we hope we have lifted most of the reviewer's reservations concerning our work.

---

## Author Response (AR1)

Dear Guillaume

Thank you for the revised manuscript and the responses to boths reviewers and to the published comment. You have taken into consideration most of the recommendations and explain why you did not wish to take on board the others. Thanks also for the revised manuscript with line numbers and clear answers that refer to the modified lines in the original article. It helps a lot.

I agree with most of your choices, but would like you to reconsider these few minor points:

Dear Christine,

Thank you for handling our manuscript and asking for these last modifications. Please, see below our responses. We hope, they are sufficient for the paper acceptation.

* For this type of study, sample size and representativeness are very important. Reducing the sample size also reveals seasonal variability (next point), which is not what is desired here. In the method section, you mention sampling " (30-100 mg) of the outermost layers". Does this sampling cover several seasons? Or is it carried out according to the length of the shell in order to remain within the same hydrological context? Can you clarify this point, or even add photos of the sampling, selecting those representing the extremes of sampling? After all, we're going from simple to triple in the mass sampled.

* the seasonal variability raised by Mr Fontugne may explain certain outliers. Depending on sampling methods, the sample may only record a very short period of life. This is an important and classic bias with the reduction in sampling size. Could you rethink this idea in the discussion, especially at river mouths and in areas of high seasonal variability? Your article would benefit from adding this possibility to the discussion.

We respond below to these two closely related comments.

Our shell sampling does not differ much from that commonly detailed in the litterature focused on the prebomb marine reservoir age reconstructions. For instance :

Southon, Kashgarian, Fontugne et al. (2002): "*The shells were cleaned by washing and grinding the surface. Aliquots a few mm square—large enough to cover at least one entire year of growth—were taken from near the outer (growing) edge of each shell. The carbonate was crushed to a 0.5–1 mm powder, etched with 0.1N HCl to remove 30% of each sample, and subsamples of 10–15 mg were hydrolyzed to CO2 with 85% phosphoric acid.*".

Tisnérat, Paterne, Métivier et al. (2010): "*For radiocarbon analyses, a slice of approximately 18 mg was sampled from the outer lip of the shells. The sample was mechanically cleaned by sand blasting to remove superficial contamination, reducing the weight by ca 20-30%. For most of the samples, the temporal resolution is estimated at ~2 years,*"

Reimer and McCormac (2002): "*Only the outer edges of the shells were used for dating so that shell deposited nearest to the time of death was sampled.*"

Siani, Paterne, Arnold et al. (2000): "*Shell samples (about 15 mg of carbonate) were etched with 0.5N hydrochloric acid and, after a 50% loss, were rinsed with deionized water to remove surface contaminations.*"

Compared to the other studies, our shell samples are 2 to 7 times bigger than sample sizes generally investigated in the litterature. The range of 30-100 mg we report here is due to the species studied: *Senilia senilis* has large and thick shells (up to 8 cm), whereas *Donax rugosus* is a species which is rather small and thin comparatively (up to 3 cm).

Some of the studies cited above indicate that their 15mg sampling was sufficient to cover 1 or even 2 yrs of the specimen growth and thus avoid the impact of seasonality of the results. If they are right, this should also be the case for our study. However, without carrying out  a complete sclerochronological study for each shell, it is clearly impossible to determine whether the shell samples cover entire years or a few seasons, which, at the very least, is already an average.

A full sclerochronological study for each is far beyond the scope of this study: 1) it would involve the complete destruction of each sample whereas some of them are unique and need to be handled with care and damaged as little as possible to be preserved for the next generations; 2) it would much too costly for the deseired objectives; 3) most of the similar studies do not go that far in their investigations and we believe that the readers are aware about the limitations of prebomb reservoir age reconstructions.

We feel that we already mentionned this limitation clearly station at line 585, owing to M. Fontugne's comment:

> *"Finally, we cannot fully rule out that these higher values represent some sub-annual variability of up to 200 $^{14}$C in the local marine reservoir age as evidenced elsewhere (Jones et al., 2007, 2010)."*

* Reviewer #2 also points out that the true measure of 14C is F14C and that the presentation in age 14C is a logarithmic transcription of F14C. I agree. This is also the recommendation of the 14C community (see e.g. Reimer 2004). This is all the more true as 14C ages should be issued rounded (Stuiver and Polach 1977), even if this is not followed by all 14C labs. So, to follow the community's logic while respecting long-standing practices, I recommend presenting results according to SacA-xxx F14C= xxx± xxx (540 ± 23)

We modified the presentation of the datalist accordingly.

* the descriptive list of samples is very important and should not be put off to an appendix. I agree with you. However, reading it is tedious. Can you imagine grouping the information in a table (in landscape mode)? Try it, if it doesn't work then let's stick with the list, perhaps in double columns (if this is allowed in the journal layout)? I'll leave you to consider the various options to get the reader to the end of the article. Attached a proposition of table.

Thank you for your proposition of table. We followed this template, and we modified the presentation of the datalist accordingly.

Many thanks in advance for the revised version and thanks again for considering Geochronology to publish the results of your research.

All the very best

Christine Hatté

Many thanks again for handling our paper.

Guillaume on behalf of all coauthors.

---

## Editor Decision (ED1)

| sample label | species | location | comment on the sample | 14C and 13C | collection date (AD) | Collector | Museum label |
|---|---|---|---|---|---|---|---|
| MNHN-IM-2022-4605 | Donax rugosus | Luanda, beach (8.82°S, 13.21°E) | Articulated specimen with remains of flesh and hinge ligament. | SacA-68817 F14C = 0.9364 ± 0.0021 (528 ± 18 BP) δ13C = 0.82 ‰ VPDB | May 18th 1910 | Mission Gruvel | Donax rugosus Linné, St Paul de Loanda plage, 18.V.10, Mission Gruvel |
| MNHN-IM-2022-4589 | Senilia senilis | Luanda, beach (8.78°S, 13.27°E) | Articulated specimen with well-preserved periostracum | SacA-68821 F14C = 0.9353 ± 0.0022 (538 ± 19 BP) δ13C = -0.16 ‰ VPDB | May 18th 1910 | Mission Gruvel | Arca (Senilia) senilis Linné, St Paul de Loanda, plage, Mission Gruvel, 18.V.1910 |
| MNHN-IM-2022-4605 | Donax rugosus | Luanda, beach (8.82°S, 13.21°E) | Articulated specimen with remains of flesh and hinge ligament. | SacA-68817 F14C = 0.9364 ± 0.0021 (528 ± 18 BP) δ13C = 0.82 ‰ VPDB | May 18th 1910 | Mission Gruvel | Museum label: Donax rugosus Linné, St Paul de Loanda plage, 18.V.10, Mission Gruvel |